# Mitochondrial Bioenergetic and Proteomic Phenotyping Reveals Organ-Specific Consequences of Chronic Kidney Disease in Mice

**DOI:** 10.3390/cells10123282

**Published:** 2021-11-24

**Authors:** Trace Thome, Madeline D. Coleman, Terence E. Ryan

**Affiliations:** 1Department of Applied Physiology and Kinesiology, University of Florida, Gainesville, FL 32611, USA; trthome@ufl.edu (T.T.); madelinedcoleman@gmail.com (M.D.C.); 2Center for Exercise Science, University of Florida, Gainesville, FL 32611, USA; 3Myology Institute, University of Florida, Gainesville, FL 32611, USA

**Keywords:** CKD, bioenergetics, muscle, cardiac, kidney, mitochondria, proteomics

## Abstract

Chronic kidney disease (CKD) results in reduced kidney function, uremia, and accumulation of uremic metabolites. Mitochondrial alterations have been suggested to play a role in the disease pathology within various tissues. The purpose of this study was to perform a comprehensive bioenergetic and proteomic phenotyping of mitochondria from skeletal muscle (SkM), cardiac muscle (CM), and renal tissue from mice with CKD. The 5-month-old C57BL/6J male mice were fed a casein control or adenine-supplemented diet for 6 months. CKD was confirmed by blood urea nitrogen. A mitochondrial diagnostic workflow was employed to examine respiratory function, membrane and redox potential, reactive oxygen species production, and maximal activities of matrix dehydrogenases and electron transport system (ETS) protein complexes. Additionally, tandem-mass-tag-assisted proteomic analyses were performed to uncover possible differences in mitochondrial protein abundance. CKD negatively impacted mitochondrial energy transduction (all *p* < 0.05) in SkM, CM, and renal mitochondria, when assessed at physiologically relevant cellular energy demands (ΔG_ATP_) and revealed the tissue-specific impact of CKD on mitochondrial health. Proteomic analyses indicated significant abundance changes in CM and renal mitochondria (115 and 164 proteins, *p* < 0.05), but no differences in SkM. Taken together, these findings reveal the tissue-specific impact of chronic renal insufficiency on mitochondrial health.

## 1. Introduction

Chronic kidney disease (CKD) stems from a progressive decline in renal function that results in elevated uremia and the accumulation of uremic metabolites which have deleterious effects on metabolism [1,2,3]. Over 697 million cases of CKD (all stages) were recorded in 2017, resulting in a global prevalence of 9.1%; and over the last two decades, the all age prevalence has increased ~30% [4,5]. A major problem with the growing prevalence of CKD is that there is no pharmacological cure for the disease and treatment options remain limited to dialysis or kidney transplantation. Considering the crucial role of the kidneys in maintenance of systemic homeostasis, CKD has the potential to negatively impact many organs in the body which contribute to a progressively debilitating phenotype characterized by impaired physical function, reduced exercise capacity, poor quality of life, cachexia, and impaired cognitive and motor abilities [6,7,8,9,10,11,12]. Many studies have described impaired metabolism and mitochondrial dysfunction in multiple tissue types around the body including skeletal muscle, cardiac muscle, and renal tissue [8,9,13,14,15,16,17,18,19,20,21]. These altered metabolic functions have been associated with muscle wasting, reduced muscular strength and exercise capacity, impaired growth and insulin signaling, left ventricular hypertrophy, increased cardiovascular events, metabolic acidosis, abnormal mitochondrial morphology, and increased oxidative stress [9,10,16,20,21].

Mitochondria are key regulators of energy homeostasis through oxidative phosphorylation (OXPHOS) as well as cellular redox homeostasis, calcium regulation, and apoptotic cellular signaling mechanisms, and have been identified as a potential target for uremic toxicity [22,23,24]. The systemic nature of uremia and the accumulation of uremic metabolites imply that many tissues/organs may be exposed to elevated concentrations of uremic metabolites, which could disrupt metabolic processes—especially in tissues with high levels of energy demand and high mitochondrial content. Deleterious effects on mitochondria have been previously demonstrated with uremic toxins such as indoxyl sulfate, kynurenic acid, and kynurenines [23,24,25,26]. The heart has a tremendous demand (~65 kg/day) for adenosine triphosphate (ATP) required to sustain its pumping activity, the majority of which (>90%) is by mitochondria [27]. The kidneys also have a high energy demand [27,28] due to the nature of producing ion gradients through secondary ATP-dependent transport to filter the blood and remove waste products. Skeletal muscle displays a high capacity for energy demand that can rapidly change in response to contractile activity such as exercise. Previous studies have shown that cardiac mitochondria in CKD display morphological abnormalities (swollen and damaged mitochondria, and reduced cristae density), reduced mitochondrial content, and increased presence of apoptotic signaling molecules found in the cytosol that are mitochondrial derived [19,20,21,22,29]. Similar to cardiac muscle, skeletal muscle mitochondria display significant impairments and are believed to contribute to muscular dysfunction and cachexia in the presence of CKD including impaired ATP production, increased oxidative stress, and mitochondrial structural alterations [14,23,24,25,26]. In the kidney, mitochondrial impairments are considered both a cause and a consequence of CKD. Numerous studies have demonstrated that increased mitochondrial fission, decreased fusion, decreased mitochondrial biogenesis signaling and fatty acid metabolism contribute to renal pathobiology; and importantly, that perseveration of renal mitochondrial function by increasing PGC1α abundance ameliorates renal fibrosis and preserves tubular filtration in the kidneys [13,15,16,19].

The goal of this study was to evaluate the organ-specific effects of CKD in mitochondrial health. To accomplish this goal, we performed comprehensive mitochondrial bioenergetic and proteomic phenotyping of mitochondrial isolated from skeletal muscle, cardiac muscle, and renal tissue from mice with and without CKD. Assessments included mitochondrial respiratory function, membrane potential, redox potential, reactive oxygen species production and electron leak, as well as enzymatic screening of matrix dehydrogenases and electron transport system protein complexes. Additionally, tandem-mass-tag-assisted proteomic analyses of mitochondrial isolates were performed to assess mitochondrial proteome abundance differences caused by CKD.

## 2. Materials and Methods

### 2.1. Animals

C57BL/6J male mice (Stock #000664) were obtained from The Jackson Laboratory at 5 months of age (N = 20 total mice). All rodents were housed in a temperature- (22 °C) and light-controlled (12 h light/12 h dark) room and maintained on standard chow diet (Envigo Teklad Global 18% Protein Rodent Diet 2918 irradiated pellet) with free access to food and water prior to CKD induction. All animal experiments adhered to the Guide for the Care and Use of Laboratory Animals from the Institute for Laboratory Animal Research, National Research Council, Washington, D.C., National Academy Press, 1996, and any updates. All procedures were approved by the Institutional Animal Care and Use Committee of the University of Florida.

### 2.2. Induction of Chronic Kidney Disease (CKD)

We utilized an established adenine diet model to induce CKD in mice [24,30,31,32]. Mice were assigned to a casein-based chow diet for 7 days, followed by induction of renal tubular injury by supplementing the diet with 0.2% adenine for 7 days, and were subsequently maintained on a 0.15% adenine diet for a total period of 6 months. CKD mice were placed back on the casein control diet for 2 weeks prior to euthanasia and terminal experiments. Control mice received a casein diet for the duration of this study. All food was provided ad libitum with free access to water.

### 2.3. Assessment of Kidney Function

Blood urea nitrogen (BUN) levels were assessed in plasma collected prior to sacrifice as performed previously [24,31]. Briefly, blood was collected in heparin-coated capillary tubes via a small ~1 mm tail snap. Collected blood was then centrifuged at 4000× *g* for 10 min at 4 °C. Collected plasma was assessed using a commercial kit (Arbor Assays #K024, Ann Arbor, Michigan, MI, USA) according to manufacturer instructions.

### 2.4. Renal Histology

Renal histology was performed using standard light microscopy as performed before [24]. Briefly, the left kidney was carefully dissected, and weights were obtained. The left kidney was then placed in OCT compound and frozen in liquid nitrogen-cooled isopentane for cryosectioning. The 5 µm thick longitudinal sections were cut using a cryotome (Leica CM3050) at −25 °C and collected on slides for staining. Masson’s Trichrome staining using Weigert’s iron hematoxylin was performed, and images were collected at 20× magnification using automated image capture/tiling in order to image the entire kidney section using an Evos FL2 Auto microscope (ThermoScientific, Waltham, MA, USA).

### 2.5. Mitochondrial Isolation Procedure

Skeletal muscle, cardiac muscle, and renal tissue were isolated as previously described [23,24,33]. Briefly, skeletal muscle was dissected from the gastrocnemius, quadriceps, hamstrings, triceps, pectorals, gluteus maximus, and erector spinae muscles and were placed in Buffer A (phosphate buffered saline supplemented with EDTA (10 mM), pH = 7.4). Both atria and ventricles of the heart and both kidneys were carefully dissected and placed in Buffer A as well. All tissues were trimmed and cleaned to remove connective tissue, fat, or any red blood cells. Thereafter, all tissues were minced and subjected to a 5 min incubation on ice in Buffer A supplemented with 0.025% trypsin (MilliporeSigma cat#T4799, Burlington, NJ, USA). Following trypsin digestion, all tissues were centrifuged at 200× *g* for 5 min to remove trypsin. Tissue pellets were then re-suspended in Buffer C (MOPS (50 mM), KCl (100 mM), EGTA (1 mM), MgSO4 (5 mM), bovine serum albumin (BSA; 2 g/L); pH = 7.1) and then homogenized via a glass-Teflon homogenizer (Wheaton) and subsequently centrifuged at 800× *g* for 10 min to pellet nuclei and other cellular components. The resulting supernatant was centrifuged at 10,000× *g* for 10 min to pellet mitochondria. All steps were performed at 4 °C or on ice. The mitochondrial pellet was gently washed to remove any damaged mitochondria and then re-suspended in Buffer B (MOPS (50 mM), KCl (100 mM), EGTA (1 mM), MgSO4 (5 mM); pH = 7.1) and protein concentration was determined using bicinchoninic acid protein assay (ThermoScientific #A53225, Walthham, MA, USA).

### 2.6. Measurement of Mitochondrial Oxphos Conductance

High-resolution respirometry was measured using Oroboros Oxygraph-2k (O2K) measuring oxygen consumption (*J*O_2_) at 37 °C in Buffer Z (105 mM K-MES, 30 mM KCl, 1 mM EGTA, 10 mM K_2_HPO_4_, 5 mM MgCl_2_-6H_2_O, 2.5 mg/mL BSA, pH 7.2), supplemented with either 5 mM creatine (conductance assay) or 20 mM creatine (mitochondrial integrity check). In order to determine the integrity of mitochondrial outer membrane for each isolate, we performed respirometry using 10 mM pyruvate + 2 mM malate as substrates, followed by a bolus of ADP (16 mM) and cytochrome C (10 mM). Samples with a cytochrome C response greater than 15% were excluded from this study (1 sample was excluded). To better understand how mitochondria function under physiological energy demands, a creatine kinase (CK) clamp was used to leverage the enzymatic activity of CK, which couples the interconversion of ATP and ADP to that of phosphocreatine (PCr) and free creatine. This allows a titration of the extra mitochondrial ATP/ADP ratio and thus free energy of ATP hydrolysis (ΔG_ATP_) across a range of energy demands [23,24,34,35]. The ΔG_ATP_ can plotted against the corresponding *J*O_2_ creating a linear force-flow relationship, where the slope represents the conductance throughout the entire mitochondrial energy transduction system [23,24,34]. For all assays, twenty micrograms of mitochondria were added to the O2K chamber in two milliliters of Buffer Z supplemented with ATP (5 mM), Cr (5 mM), PCr (1 mM), CK (20 U/mL). Mitochondrial were fueled with a mixture of carbohydrate and fatty acid (FA) substrates (5 mM pyruvate + 2.5 mM malate + 0.2 mM octanoyl-carnitine).

### 2.7. Mitochondrial Hydrogen Peroxide (H_2_O_2_) Production and Electron Leak

Mitochondrial H_2_O_2_ production was measured fluorometrically via Amplex Ultra Red (AUR)/horseradish peroxidase (HRP) detection system (Ex/Em 530/590 nm) as described previously [23,24,34]. Fluorescence was captured via a QuantaMaster Spectrofluormeter (Horiba Scientific, Piscataway, NJ, USA) and for each experiment, resorufin fluorescence was converted to pmols of H_2_O_2_ via a H_2_O_2_ standard curve in identical substrate conditions as the experimental protocol. All experiments were performed at 37 °C in a 0.75 mL reaction volume. Buffer for all assays was identical to OXPHOS conductance experiments as well as the utilization of fuel sources (pyruvate, malate, and FA).

### 2.8. NAD(P)H/NAD(P)^+^ Redox Potential

Fluorescent determination of NAD(P)H/NAD(P)^+^ was carried out using a QuantaMaster Spectrofluormeter (Horiba Scientific, Piscataway, USA) [35,36]. To measure the NAD(P)H/NAD(P)^+^ redox potential, NAD(P)^+^ auto fluorescence was measured kinetically via excitation/emission at 340/450 nm. All experiments were carried out at 37 °C in 0.75 mL of Buffer Z supplemented with 5 mM of creatine, 1 mM phosphocreatine (PCr), and 20 U/mL creatine kinase (CK). Initially, 75 µg of mitochondria was added to the assay buffer to determine a baseline measurement (0% reduction) followed by the addition of respiratory substrates (5 mM pyruvate, 2.5 mM malate, and 0.2 mM octanoyl-carnitine) inducing state 2 respiration. Following state 2 respiration, adenosine triphosphate (5 mM; ATP) was added, then sequential titrations of PCr (6 mM, 15 mM, and 30 mM) were added utilizing the CK clamp to titrate the extra mitochondrial ATP/ADP ratio. Last, 4 mM potassium cyanide (sigma#60178) was added to induce full reduction (100%) within the NAD(P)H/NAD(P)+ couple. The fluorescence of 340/450 nm was collected through the whole experiment allowing us to use the presence of mitochondria alone with no substrates as 0% reduction and the addition of cyanide as 100% reduction. The results were then expressed as percentage reduction using the formula: % reduction = (X − X_0%_red)/(X_100%_red − X_0%_red).

### 2.9. Mitochondrial Membrane Potential (Δψ)

Fluorescent determination of Δψ was performed using a QuantaMaster Spectrofluormeter (Horiba Scientific, Piscataway, USA) as performed previously [34,36] by taking the Tetramethylrhodamine, methyl ester (TMRM) fluorescence ratio of excitation emission parameters [Ex/Em = (572/590 nm)/(551/590 nm)]. All experiments were carried out at 37 °C in 0.75 mL of Buffer Z supplemented with 5 mM of creatine, 1 mM phosphocreatine (PCr), 20 U/mL creatine kinase (CK), and 0.2 µM TMRM. Briefly, 75 µg of mitochondria was added to the assay buffer to determine a baseline measurement followed by the addition of respiratory substrates (5 mM pyruvate, 2.5 mM malate, and 0.2 mM octanoyl-carnitine) inducing state 2 respiration. Following state 2 respiration, adenosine triphosphate (5 mM; ATP) was added, then sequential titrations of PCr (6 mM, 15 mM, and 30 mM) were added. The fluorescence excitation ratio collected was then converted to millivolts using a KCl standard curve carried out in the presence of valinomycin, a potassium ionophore [34,37].

### 2.10. Assessment of Mitochondrial Dehydrogenase Activity

The activity of several mitochondrial matrix dehydrogenases (*J*NAD(P)H) were measured fluorometrically using NAD(P)H auto fluorescence (Ex/Em = 340/450 nm) in a 96-well plate read kinetically on a BioTek Synergy 2 Multimode Microplate Reader as performed previously [23,24]. For all assays, buffer Z was supplemented with alamethicin (0.03 mg/mL), rotenone (0.005 mM), NAD+ or NADP+ (2 mM). For pyruvate dehydrogenase, alpha ketoglutarate dehydrogenase, and branched chain ketoacid dehydrogenase, the assay buffer was supplemented with cofactors Coenzyme A (0.1 mM), and thiamine pyrophosphate (0.3 mM). For each dehydrogenase, assay buffer was loaded and warmed to 37 °C followed by the addition of mitochondria. Mitochondria were permeabilized by alamethicin for five minutes. Dehydrogenase activity was then initiated by addition of the following substrates run in parallel: pyruvate (5 mM), malate (5 mM), glutamate (10 mM), alpha ketoglutarate (10 mM), branched-keto-acid-methylvalerate (5 mM), or isocitrate (5 mM). Rates of NADH production was calculated as the slope of linear portions of the NADH curves. Fluorescence values were converted to pmoles of NAD(P)H/NAD(P)H via a standard curve.

### 2.11. Hydroxyacyl-CoA Dehydrogenase Activity

Isolated mitochondrial lysates were prepared by addition of Cell Lytic M (MilliporeSigma #C2978, Burlington, USA) at a protein concentration of 1 µg/µL. Dehydrogenase activity was measured fluorometrically using NAD(P)H auto fluorescence (Ex/Em = 340/450 nm) in a 96-well plate read kinetically on a BioTek Synergy 2 Multimode Microplate Reader as performed previously [34]. All experiments were performed at 37 °C in 200 µL of Buffer E (HEPES (20 mM), KCl (100 mM), KH2PO4 (2.5 mM), MgCl2 6H2O (2.5 mM), and 1% glycerol) supplemented with 0.005 mM Rotenone and 0.2 mM NADH. Five micrograms of mitochondria were added to each well, and the experiment was initiated by the addition of 0.2 mM acetoacetyl-CoA. Dehydrogenase activity was determined following the degradation of the NADH to NAD+. Fluorescence values were then converted to pmols of NADH via a standard curve.

### 2.12. ATP Synthase Activity (Complex V)

ATP synthase activity (complex V) was measured as previously described [23,24]. Briefly, mitochondria were incubated with Cell lytic M (MilliporeSigma #C2978, Burlington, USA) to lyse mitochondria. Buffer E (HEPES (20 mM), KCl (100 mM), KH2PO4 (2.5 mM), MgCl2 6H2O (2.5 mM), and 1% glycerol) was supplemented with lactate dehydrogenase (10 mM), pyruvate kinase (10 mM), rotenone (0.005 mM), phosphoenol-pyruvate (PEP, 5 mM), and NADH (0.2 mM). NADH levels were determined via auto fluorescence (Ex/Em = 340/450 nm). For this assay, the ATP synthase functions to hydrolyze ATP because lysis of the mitochondria will dissipate their ability to establish a membrane potential (which normally drives ATP synthesis). Using a pyruvate kinase/lactate dehydrogenase coupled assay, ATP hydrolysis by the ATP synthase is coupled to NADH consumption in a 1:1 stoichiometry. Thus, the rate of decrease in NADH auto fluorescence can be used as a measure maximal ATP synthase activity. Fluorescence values were converted to pmoles of NADH via a standard curve.

### 2.13. Mitochondrial OXPHOS Complex Activities

The enzyme activity of all electron transport system (ETS) complexes was determined spectrophotometrically as previously described [24,38].

### 2.14. Tmt-Assisted Proteomics

One hundred micrograms of isolated mitochondria were used per sample per tissue type (n = 3/group/tissue). Mitochondria were spun down at 10,000× *g* for 10 min at 4 °C and mitochondrial isolation medium was removed and mitochondria were resuspended in 150 µL of Chaps lysis buffer (150 mM KCl, 50 mM HEPES, 0.1% chaps (ThermoScientific #28300, Waltham, MA, USA), and protease inhibitors (ThermoScientific #A32955, Waltham, MA, USA). Samples were subjected to hand homogenization in glass homogenizers and three freeze/thaw cycles in liquid nitrogen and vortexed vigorously, followed by further disruption via sonication with a probe sonicator in three 5 s bursts (30% amplitude) on ice. The samples were then spun down at 16,000× *g* for 10 min at 4 °C. The supernatant was collected, and protein concentration was determined using a bicinchoninic acid protein assay (ThermoScientific #A53225, Waltham, MA, USA) and volume was brought to 100 µL total per sample using 100 mM TEAB (ThermoScientific #90114, Waltham, MA, USA) with a total of 100 µg per sample. The samples were then reduced with the addition of 5 µL of 200 mM TCEP and incubated at 55 °C for 1 h. Following reduction, samples were alkylated via the addition of 5 µL of 375 mM iodoacetamide and incubated for 30 min at room temperature protected from light. Protein samples were then precipitated by adding 400 µL of methanol, vortexed to mix thoroughly, and centrifuged at 9000× *g* for 10 s at room temperature. A volume of 100 µL of chloroform was then added and vortexed followed by another 9000× *g* spin for 10 s. A volume of 300 µL of water (LC–MS grade ThermoScientific #51101, Waltham, MA, USA) was added to the samples, vortexed vigorously and centrifuged for 1 min at 9000× *g* at room temperature. This step has 3 phases. The top layer (supernatant = mixture of water and methanol) is removed carefully and 300 µL of methanol is added and mixed vigorously and centrifuged at 9000× *g* for 2 min at room temperature. The supernatant was then removed, and remaining fluid was gently aspirated using a vacuum concentrator until the pellet was left a little moist. Precipitated proteins were then resuspended in 100 µL of 100 mM TEAB and 2.5 µg of trypsin (ThermoScientific #A0006, Waltham, MA, USA) was added per 100 µg of protein and digested overnight at 37 °C. Following overnight protein digestion, samples were labeled using 11-plex tandem-mass-tag (TMT) reagents (0.8 mg dissolved in 41 µL of acetonitrile LC–MS grade (ThermoScientific # 51101, Waltham, MA, USA) and incubated for 1 h at room temperature. After incubation, the reaction was quenched with the addition of 8 µL of 5% hydroxylamine and incubated for 15 min. Labeled peptides were desalted with C18-solid phase extraction and dissolved in strong cation exchange (SCX) solvent A (25% (*v/v*) acetonitrile, 10 mM ammonium formate, and 0.1% (*v*/*v*) formic acid, pH 2.8). The peptides were fractionated using an Agilent HPLC 1260 with a polysulfoethyl A column (2.1 × 100 mm, 5 µm, 300 Å; PolyLC, Columbia, MD, USA). Peptides were eluted with a linear gradient of 0–20% solvent B (25% (*v*/*v*) acetonitrile and 500 mM ammonium formate, pH 6.8) over 50 min followed by ramping up to 100% solvent B in 5 min. The absorbance at 280 nm was monitored and a total of 16 fractions were collected. The fractions were lyophilized, desalted, and resuspended in LC solvent A (0.1% formic acid in 99.9% water (*v*/*v*)). A hybrid quadrupole Orbitrap (Q Exactive Plus) MS system (ThermoScientific, Walthham, MA, USA) was used with high energy collision dissociation (HCD) in each MS and MS/MS cycle. The MS system was interfaced with an automated Easy-nLC 1000 system (ThermoScientific, Walthham, MA, USA). Each sample fraction was loaded onto an Acclaim Pepmap 100 pre-column (20 mm × 75 μm; 3 μm C18) and separated on a PepMap RSLC analytical column (250 mm × 75 μm; 2 μm C18) at a flow rate at 350 nL/min during a linear gradient from solvent A (0.1% formic acid (*v*/*v*)) to 30% solvent B (0.1% formic acid (*v*/*v*) and 99.9% acetonitrile (*v*/*v*)) for 95 min, to 98% solvent B for 15 min, and hold 98% solvent B for additional 30 min. Full MS scans were acquired in the Orbitrap mass analyzer over m/z 400–2000 range with resolution 70,000 at 200 m/z. The top ten most intense peaks with charge state ≥3 was fragmented in the HCD collision cell normalized collision energy of 28%, (the isolation window was 0.7 m/z). The maximum ion injection times for the survey scan and the MS/MS scans were 250 ms, respectively, and the ion target values were set to 3 × 10^6^ and 1 × 10^6^, respectively. Selected sequenced ions were dynamically excluded for 60 s. Proteomics data have been deposited to the jPOST repository (ID JPST000977) and Proteome Xchange (ID PXD021847).

### 2.15. Data Analysis for Mitochondrial Proteomics

Raw MS/MS data files were processed against Uniprot Mus musculus reference proteome database using the Proteome Discoverer v2.4 (ThermoScientific, Waltham, MA, USA). The following parameters were used: peptide tolerance at 10 ppm, tandem MS tolerance at ±0.02 Da, peptide charges of 2+ to 5+, trypsin as the enzyme, allowing one missed cleavage, TMT label (229.163 Da on-peptide N-term and K) and carbamidomethyl (C) as fixed modifications, oxidation (15.995 Da on M) as a variable modification. The false discovery rate (FDR) was calculated using Percolator algorithm in the Proteome Discoverer workflow based on the search results against a decoy database and was set at 1% FDR. For protein quantification, only MS/MS spectra that were unique to a particular protein and where the sum of the signal-to-noise ratios for all the peak pairs >9 was used for quantification. M2 reporter (TMT) intensities were summed together for each TMT channel, each channel’s sum was divided by the average of all channels’ sums, resulting in channel-specific loading control normalization factors to correct for any deviation from equal protein input in the 11-plex experiments. Reporter intensities were divided by the loading control normalization factors for each respective TMT channel. All loading control-normalized reporter intensities were converted to log_2_ space and the average value from the combined samples was subtracted from each sample-specific measurement to normalize the relative measurements to the mean. For each comparison, protein abundances were analyzed for group average, standard deviation, two-tailed Student’s t-test (equal variance), and a Benjamini–Hochberg [39] adjusted *p*-value.

### 2.16. Statistical Analysis

Data are presented as the mean ± SD. Normality of data was assessed using the Shapiro–Wilk test. Data were first analyzed using two-tailed unpaired *t*-test. Pearson correlations were performed using two-tailed statistical testing. All statistical analysis was performed in GraphPad Prism (Version 8.4). In all cases, *p* < 0.05 was considered statistically significant. All data can be found at the following DOI: 10.6084/m9.figshare.16837924.

## 3. Results

### 3.1. Mitochondrial Phenotyping Identifies Organ-Specific Bioenergetic Profiles

Although each of our cells contain the same nuclear and mitochondrial genomes, mitochondrial metabolism is not uniform across cell types and instead is finely matched to the energetic demands of the cell. This statement infers that not all mitochondria are created equal but are rather equipped to handle specific cellular stresses in which they reside. For instance, skeletal muscle, cardiac muscle, and renal tissue were targeted for bioenergetic assessments in this study due to their high metabolic demand for energy (ATP) required for cellular tasks such as contraction (skeletal and cardiac muscle) or production of ion gradients utilized for reabsorption or excretion of metabolites (kidneys). Given the diversity of cellular function, we first wanted to establish differences in mitochondrial energy transduction through these three highly metabolic tissues utilizing a rigorously developed mitochondrial energy transduction phenotyping platform [34,35]. Using combined carbohydrate (pyruvate and malate) and fatty acid (FA, octanoyl-L-Carnitine) fuel sources (Figure 1A), cardiac mitochondria proved to have the highest rate of respiration at the highest energy demand (Figure 1B) while also displaying the greatest OXPHOS conductance (Figure 1C). Interestingly, the kidney’s OXPHOS conductance was significantly lower than muscle and cardiac mitochondria (~82% lower than cardiac mitochondria, and ~72% lower than skeletal muscle mitochondria). Following the initial OXPHOS conductance assessment, we assessed the redox potential across energy demands which displayed energetic differences across tissue types as well. At lower energy demands (ΔG_ATP_ = −15.24 and −14.72 Kcal/mol) the kidney exhibited a more reduced redox state (higher levels of NAD(P)H/NAD(P)+) compared to skeletal muscle and cardiac muscle mitochondria (Figure 1D). However, at the higher energy demands (ΔG_ATP_ = −12.94 and −14.18 Kcal/mol), the kidney was only significantly higher than cardiac muscle mitochondria (*p* < 0.01, Figure 1D)). Moreover, mitochondrial membrane potential was significantly different across tissues for each level of energy demand (Figure 1E). The relationship between *J*O_2_ and membrane potential further highlights the diversity in mitochondrial energetics across tissues (Figure 1F). Intriguingly, mitochondrial electron leak and H_2_O_2_ production were significantly greater in kidney mitochondria compared to skeletal or cardiac muscle mitochondria at all energy demand levels (Figure 1G,H). At the highest energy demand (ΔG_ATP_ = −12.94 Kcal/mol), skeletal muscle had higher estimated electron leak than cardiac muscle mitochondria; however, there was no significant difference between skeletal and cardiac muscle H_2_O_2_ production (Figure 1G,H). Lastly, individual enzyme assays demonstrate a diverse landscape of matrix dehydrogenases and ETS enzyme function across tissue types (Figure 1I,J).

### 3.2. Long-Term Adenine Feeding Causes Severe CKD

Supplementation of diet with adenine is an established model for inducing CKD in rodent [25,31,32,33]. The adenine consumption results in excessive production and subsequent crystallization of 2,8-dihydroxyadenine in the renal tubules causing tubulointerstitial nephropathy and consequently kidney damage [30]. The adenine diet model has also been shown to induce significant uremia [24,31,40]. While most studies utilizing the adenine diet model in mice do not exceed the length of 3–4 months of induction, the present study used fully mature 5-month-old mice which a received casein (control) or adenine-supplemented diet (CKD) treatment for 24 weeks (~6 months). The longer duration was implemented to provide a chronic and severe phenotype associated with CKD. Furthermore, CKD is a wasting disease usually characterized by reduced body and kidney weight, elevated levels of blood urea nitrogen (BUN), appearance of cyst like structures, and increased kidney fibrosis and inflammation [1,2,6]. Six months of adenine feeding resulted in significantly lower body weights (32.9 ± 1.2 g vs. 23.6 ± 1.3 g, *p* < 0.0001) and kidney weights (178.8 ± 27.7 mg vs. 137.7 ± 24.4 mg, *p* < 0.05) (Figure 2A–C) compared to casein fed control mice. Long-term administration of dietary adenine also resulted in renal morphological changes including large cyst-like structures, deposition of crystals, and kidney fibrosis (Figure 2D,F). However, we were unable to quantify morphological changes to the kidneys, such as fibrotic area, due to the tissue required for mitochondrial functional assessments, but a representative histological image of a control vs. CKD kidney using Masson’s trichrome staining is shown in Figure 2F. To confirm kidney damage and filtering dysfunction, the BUN levels in the plasma were assessed. Adenine-induced CKD resulted in large increases in BUN compared to the casein control group (24.14 ± 7.70 mg/dL vs. 105.3 ± 18.35 mg/dL, *p* < 0.0001) (Figure 2E). Notably, two adenine-fed mice unexpectedly died between the 5th and 6th months on adenine diet. The cause of death is unknown as necroscopy was not performed, but this may be related to the severity of kidney injury or another CKD-associated pathology contributing to mortality.

### 3.3. Chronic Kidney Disease Impairs Mitochondrial Energy Transduction in Skeletal Muscle, Cardiac Muscle, and Renal Tissue

To understand how CKD impacts mitochondrial energy transduction, we employed the mitochondrial phenotyping platform described earlier (Figure 1A). Supported by a mixture of carbohydrate and fatty acids (FA), CKD mice displayed significant reductions in OXPHOS conductance in skeletal muscle (~20% decrease in OXPHOS conductance, *p* < 0.05) and cardiac muscle mitochondria (~30% decrease in OXPHOS conductance, *p* < 0.001) (Figure 3A,B). Not surprisingly, renal mitochondria from CKD mice had lower oxygen consumption rates at all levels of energy demand and almost no ability to increase OXPHOS in response to ΔG_ATP_, evidenced by a ~75% reduction in respiratory conductance (Figure 3C).

### 3.4. Tissue-Specific Changes in Mitochondrial Redox and Membrane Potential in CKD

In parallel to respiratory function, we performed analysis of mitochondrial redox potential (NAD(P)H/NAD(P)+) and membrane potential (Δψ) across the range of ΔG_ATP_. Skeletal muscle mitochondria from CKD mice exhibited a more oxidized NAD(P)H/NAD(P)^+^ redox potential at all energy demands (Figure 3D, *p* < 0.001 at all energy demands); however, the Δψ was not significantly different between groups even though there appeared to be a leftward shift of the membrane potential curve when plotted against *J*O_2_ (reduced Δψ at each energy demand) (Figure 3E–G). In skeletal muscle mitochondria, a more oxidized redox potential is observed and could indicate that production of NAD(P)H through matrix dehydrogenases is impaired in CKD mice. This presumably reduces the electrons supply to the ETS proteins which ultimately convert the redox potential to the Δψ (or proton motive force). Similar to skeletal muscle, cardiac mitochondrial redox potential in CKD mice was significantly more oxidized at lower energy demands; however, significance was not found at higher levels of energy demand (Figure 3D). Moreover, cardiac mitochondria from CKD mice tended to have lower mitochondrial membrane potentials compared to non-CKD control mice (Figure 3E–G), although these differences were not statistically significant (*p* = 0.08, *p* = 0.08, *p* = 0.10, *p* = 0.19 in order from lowest ΔG_ATP_ to the highest). Although statistically insignificant, small changes in Δψ could implicate an impairment in the ability to pump protons into the inner membrane space via ETS enzymes.

Consistent with the large impairment in respiratory conductance, renal mitochondria from CKD mice exhibited large alterations in both redox and Δψ compared to control mice. CKD renal mitochondria presented with a hyper-reduced redox state at lower energy demands coupled to a significant decrease in mitochondrial Δψ at all ΔG_ATP_ values (Figure 3D,E). With these observations in mind, the matrix dehydrogenase enzymes producing the redox potential are not likely to be the main source of impaired energy transduction. Coupled with the hyper-reduced redox state, the observation of lower mitochondrial Δψ is indicative of impaired enzymatic function of the ETS complexes that convert the redox potential into the proton motive force. Furthermore, when Δψ is plotted versus *J*O_2_ (Figure 3G), the change in membrane potential is practically unresponsive to energy demand alteration.

### 3.5. H_2_O_2_ Production and Electron Leak Are Elevated in Cardiac but Not Skeletal Muscle or Kidney Mitochondria in CKD

As described, mitochondrial OXPHOS conductance was reduced in all three highly energetic tissues in the presence of CKD. A reduction in conductance can infer an increase in resistance of energy transduction, which could manifest as elevated electron leak/reactive oxygen species (ROS) production at numerous sites within the mitochondria. Intriguingly, oxidative stress has been postulated to play a significant role in the pathology and progression of CKD-induced cellular stress in skeletal, cardiac, and renal tissue [10,11,12,15,19,20]. Thus, we assessed mitochondrial H_2_O_2_ production and electron leak (%) in skeletal muscle, cardiac muscle, and renal mitochondria in severe CKD mice. Using the Amplex UltraRed combination with the CK clamp system, we were able to measure H_2_O_2_ production at each level of physiologically relevant energy demand and estimate electron leak by dividing the *J*H_2_O_2_ by the corresponding *J*O_2_ under identical conditions. Consistent with a previous report [24], skeletal muscle mitochondria showed no increase in H_2_O_2_ production or electron leak (Figure 3F–G) under relevant levels of energy demand. Cardiac mitochondria from CKD mice displayed increased H_2_O_2_ production at the lower (resting or near-resting) energy demands (ΔG_ATP_ = −14.72 and −15.24 kcal/mol) and non-significant increases at the higher energy demands (ΔG_ATP_ = −12.94 and −14.18 kcal/mol) (Figure 3F). Interestingly, the estimated electron leak in CKD cardiac mitochondria was significantly increased at all four energy demands (Figure 3G). In stark contrast, renal mitochondria form CKD animals displayed marked reductions in H_2_O_2_ production at all energy demands (Figure 3F, *p* < 0.001). When analyzed as percentage electron leak at each energy demand, there were no statistically significant differences between control and CKD (Figure 3G).

### 3.6. Mitochondrial Matrix Enzyme Activities and Production of NADH Are Differentially Altered across Tissues

Next, we performed biochemical screening for matrix dehydrogenase enzyme activities in all three tissues. As predicted above, results of this screening demonstrated significant impairments in skeletal muscle dehydrogenase enzyme activity compared to cardiac muscle mitochondria and renal mitochondria. In skeletal muscle, alpha-ketoglutarate dehydrogenase (AKGDH) displayed a significant reduction in the enzyme activity (*p* < 0.05), while other dehydrogenases demonstrated insignificant reductions in activity (Figure 4A). These included pyruvate dehydrogenase (PDH, *p* = 0.080), malic enzyme (ME, *p* = 0.092), and isocitrate dehydrogenase (ICDH, *p* = 0.071) (Figure 4A). Considering the unexpected death of mice at the latter stages of the intervention, this study may have been underpowered to detect the impairments in these other dehydrogenases in skeletal muscle. Similar to skeletal muscle, cardiac mitochondria from CKD mice exhibited reduced AKGDH activity (*p* < 0.01) (Figure 4B) compared to controls. Conversely, the kidney displayed a hyper-reduced state, which would suggest dehydrogenase activity was normal. Indeed, this prediction was proved true demonstrating normal, and in some cases increased dehydrogenase activity in CKD mice (BCKDH, *p* = 0.051 vs. control) (Figure 4C).

### 3.7. CKD Impairs Electron Transport System Enzyme Activity in a Tissue-Specific Manner

Following assessment of mitochondrial redox potential and matrix dehydrogenases, we next assessed whether the enzymes of the ETS were impacted by the CKD milieu. Figure 5 shows the activities of each individual complex within the ETS across each tissue type. Skeletal muscle demonstrated a ~20% and ~38% reduction in complex III and IV activities, respectively (*p* < 0.05 for both), and a non-significant reduction (~34%) in complex I activity (*p* = 0.068) (Figure 5A). In contrast, cardiac mitochondria had significant reductions in the activity of complex IV only (~36%, and *p* < 0.05) (Figure 5B). Renal mitochondria displayed a significantly reduced activity of complex II (~24%), III (~30%), and IV (~43%) compared to controls (all *p* < 0.05) (Figure 5C). In contrast, complex V (also known as the ATP synthase) of the ETS was not altered by CKD in skeletal muscle or renal mitochondria (Figure 5A,C) and was slightly increased in cardiac mitochondria from CKD animals (*p* = 0.051, Figure 5B).

### 3.8. Tissue-Specific Mitochondrial Proteomes in CKD

To determine if the functional changes observed in CKD mitochondrial were related to changes in protein abundance, we performed TMT-assisted proteomics analyses on isolated mitochondrial samples (n = 3 per group, Figure 6A). Volcano plots of protein abundance changes for each tissue are shown in Figure 6B. Notably, the skeletal muscle proteome abundance was nearly identical in control and CKD mice, with no statistically different proteins after correction for multiple comparisons (Benjamini–Hochberg adjusted) whereas significant alterations were observed in both cardiac and renal CKD mitochondria (Figure 6B,C). Predictably, the renal mitochondrial proteome exhibited the highest number of differentially abundant proteins (Figure 6C). Notably, cardiac mitochondria displayed 115 differentially abundant proteins in CKD animals (Figure 6C). Interestingly, in both renal and cardiac mitochondria, the majority (>70%) of differentially abundant proteins were upregulated compared to non-CKD controls. Venn diagrams presented in Figure 6 display the number of common and differential protein changes each tissue. A complete list of proteome data is provided at DOI: 10.6084/m9.figshare.16837924. Geno ontology analysis of identified proteins confirmed that highly prevalent biological processes involved in mitochondrial metabolism/biology, as expected considering mitochondrial isolates were used for TMT labeling (Figure 6D).

To assess whether alterations in protein abundance were associated with mitochondrial OXPHOS conductance, Pearson correlation analyses were performed to identify proteins that were significantly correlated with OXPHOS conductance. Interestingly, skeletal muscle displayed zero statistically different proteins in CKD and control mitochondrial proteomes; however, four proteins (CYB5R1, IDH3g, NDUFS7, and RMDN1) were found to be significantly correlated with OXPHOS conductance (*p* < 0.05, Figure 7A,D). Cardiac mitochondria exhibited more diverse associations with 134 significant proteins related to OXPHPOS conductance. Two of those proteins (CYB5R1 and NDUFS7) were also correlated to OXPHOS in skeletal muscle (Figure 7B–D). There were 210 proteins in renal mitochondria that correlated with OXPHOS conductance, two of those proteins were also related to skeletal muscle and 77 proteins were similar to cardiac mitochondria (Figure 7D). Only a single protein, CYB5R1 (NADH-cytochrome b5 reductase 1), was found to significantly correlate with OXPHOS conductance in all three tissues.

## 4. Discussion

Altered mitochondrial function has been implicated to play a pathologic role in renal and cardiac fibrosis, left ventricular hypertrophy, inflammation, apoptosis, energy imbalance, muscle wasting, and reduced exercise capacity in the context of CKD. However, the tissue/organ-specific metabolic disturbances associated with CKD are not fully understood. In this study, a comprehensive mitochondrial phenotyping approach was applied to mice subjected to chronic adenine feeding or a control diet. Adenine-induced CKD was found to negatively impact mitochondrial OXPHOS in skeletal muscle (20% decrease), cardiac muscle (30% decrease), and the kidney (75% decrease). To identify potential sites of dysfunction, redox potential (NAD(P)H/NAD(P)^+^), membrane potential (Δψ), enzyme activities, and oxidative stress were assessed. Skeletal muscle mitochondria demonstrated a more oxidized redox state and modest reductions in Δψ which were confirmed to be driven by impaired matrix dehydrogenase enzyme activities (PDH, AKGDH, ME, and ICDH) and significant reductions in complex III and IV activities. Notably, these impairments in CKD skeletal muscle mitochondria did not cause increases in H_2_O_2_ production or electron leak. Cardiac muscle mitochondria displayed a similar oxidized redox potential and modest decrease Δψ, which were likely driven by the reduced activity of matrix dehydrogenase AKGDH and complex IV (cytochrome c oxidase). The large (75%) reduction in renal OXPHOS conductance, was paralleled by a hyper-reduced redox potential, a lower Δψ, which appear to have been caused by normal, or in some cases elevated, matrix dehydrogenase activity, and significant reductions in the activity of complexes II, III, and IV of the mitochondrial ETS. Additional proteomics profiling of all three mitochondrial proteomes uncovered tissue-specific changes in mitochondrial protein abundance caused by CKD.

Mitochondrial dysfunction has been postulated to play a role in the myopathic phenotype of CKD including muscle atrophy, weakness, and exercise intolerance. In the present, we confirm a previous report documenting muscle mitochondrial impairments in OXPHOS are driven by dehydrogenase enzymes and not robust changes in mitochondrial content or oxidative stress [24]. Interestingly, the current study, which involved longer duration of adenine feeding (24 weeks vs. 10 weeks), did not appear to result in a progressive decline in OXPHOS conductance (~20% decrease in both studies). Nonetheless, the source of impairment (AKGDH and PDH) was consistent between both studies. Analysis of the mitochondrial proteome in skeletal muscle did not find differences in protein abundances of dehydrogenases in CKD, suggesting that other factors such as post-translational modifications must be responsible for the decreased enzymatic activity. In addition to dehydrogenase impairments, this study reported a lower complex III and IV activity of the skeletal muscle mitochondria from CKD mice. These findings are consistent with two previous studies examining rats subject to a 5/6th nephrectomy CKD model: Yazdi et al. [41], which reported reduced complex IV activity in skeletal muscle mitochondria; and Wang et al. [42], which showed significantly reduced complex I, II, III and IV activities. In contrast to several studies that have reported decrease mitochondrial content via immunoblotting [10,43,44], the current study did not detect changes in abundance of subunits of mitochondrial ETS utilizing TMT-assisted proteomic analyses.

Heart disease is a common complication of CKD and approximately 30% of CKD patients suffer from heart failure [20]. ATP turnover is very high in the heart and continuous supply of ATP is required to maintain its mechanical workload. The principal source of this ATP is oxidative phosphorylation in the mitochondrion, and it has been well documented that cardiac metabolic remodeling is a key contributor to the impaired contractile activity of the heart [45,46]. In the present study, CKD was found to impair cardiac mitochondrial function by ~30%, an observation that was especially evident at higher levels of energy demand. Similarly, in rats with nephrectomies, cardiac mitochondrial respiratory control ratios were decreased [23]. Unfortunately, functional measurements of the heart were not possible in the current study, so future studies are needed to determine if this level of impairment in cardiac oxidative metabolism is linked to altered heart function in CKD mice. Consequent to impaired oxidative function, CKD heart mitochondria notably displayed higher levels of H_2_O_2_ production at ‘resting’ levels of energy demand, while an estimation of electron leak was significantly increased at all ΔG_ATP_ levels tested. Oxidative stress has also been implicated to play a major role in pathological cardiac remodeling including cardiac hypertrophy, interstitial fibrosis, cardiac senescence, as well as metabolic aging [21,22,47]. To date, there has been limited investigation into the impact of the CKD milieu on cardiac metabolism. Nonetheless, there are several studies that have reported observations related to the current work. For example, isolated cardiac myocytes from C57BL6J mice following acute kidney injury (AKI) demonstrated an increase in ROS production utilizing a ROS-based fluorescent probe (DCFHA-DA) [48]; and in another study in rats with nephrectomies, cardiomyocytes displayed increased vulnerability to oxidant-induced death [22]. It is believed that increased ROS production within the mitochondria can cause oxidative post-translational modifications to respiratory proteins and mtDNA that cumulatively impair the organelle’s function. In support of this idea, transgenic mice that express mitochondrial targeted catalase display resistance to cardiac remodeling with aging [49,50]. Alterations in mitochondrial dynamics such as mitochondrial fragmentation, cristae swelling, and reduced total mitochondrial content have also been documented in the heart of animals with AKI or CKD [19,20,21,22]. Further to this, ROS and oxidative stress have been linked to these outcomes [51]. Taken together, the current study builds upon the existing literature and documents the biochemical remodeling of cardiac mitochondrial energetics in mice with CKD.

The kidney has a high energy demand and renal mitochondrial function has been a consistently growing area of pre-clinical and clinical research across a wide range of renal pathologies, including CKD and AKI [13,15,28]. In the current study, we report a ~75% decrease in mitochondrial OXPHOS function driven primarily by impairments in the ETS protein complexes. Considering that mitochondrial ATP production is the primary energy source that powers the waste filtration and fluid/electrolyte homeostasis functions within the kidney, this deficit undoubtedly contributes to the poor renal function caused by adenine feeding. The findings herein are in agreement with the body of literature demonstrating the importance of renal mitochondrial health in determining the organs function across a range of renal pathologies (reviewed in [28]). Importantly, numerous studies have demonstrated that therapeutic improvement in renal mitochondrial function can promote functional recovery of the organ and reduce fibrosis [51,52,53,54,55]. Additional aspects of mitochondrial structure/function also support a critical role of the organelle in kidney function. For instance, mitochondrial morphology and dynamics are altered in CKD, resulting in fragmented and swollen mitochondria with increased fission signaling (DRP1), decreased fusion signaling (OPA1 an MFN2), and impaired autophagy [13,15,18]. Mitochondrial biogenesis is also deficient in CKD kidneys and reduced PGC1α signaling has been linked to renal impairments in lipid oxidation [13,16,28]. Further to this, the present study also reports a marked alteration of the renal mitochondrial proteome (129 upregulated and 35 down regulated mitochondrial proteins in CKD, Figure 6C). Taken together, these findings add to the growing body of literature on renal mitochondrial pathologies and identifies a source of OXPHOS impairment as disruptions in ETS protein complex function.

Although this study employed a wide-ranging assessment of mitochondrial energetics in CKD across several tissues, there are some notable limitations. First, this study was performed only on fully mature male mice, as experiments with females were disrupted by the COVID-19 pandemic. There have been reports of sex differences in CKD-associated pathologies in C57BL6/J mice subjected to the adenine diet model [24,56,57]. Because female mice were not studied, extrapolations from the current findings are not recommended. Second, a combination of carbohydrate (pyruvate and malate) and FA (octanoyl-L-carnitine) fuel sources were provided during these assessments; however, in vivo metabolism involves numerous other fuel sources (amino acids, ketones, etc.). Third, proteomic experiments only quantified the abundance of protein, but did not consider post-translational modifications which are known to alter protein/enzyme function [58,59,60]. For example, phosphorylation is a well-known reversible modification that is utilized to increase/decrease enzyme function in response to the metabolic stress within the mitochondria and ~91% of mitochondrial proteins have at least one reported phosphorylation site [58,61]. Hyper-acetylation/succinylation has also been described to play a role in disease pathologies such as heart failure and obesity and would be necessary to determine even though hyper-acetylation/succinylation has recently been shown to display unremarkable impairments to mitochondrial OXPHOS function [62]. Lastly, the procedure for isolating mitochondria from tissues/organs can result in a potential selection bias, such that the healthiest organelles survival the procedure. Considering this, some damaged mitochondria within the tissues of CKD mice may have been lost during isolation and thus have not contributed to the results presented herein. If this is the case, one would reasonably hypothesize that the mitochondrial affect caused by CKD is greater than the reported results in this study.

## 5. Conclusions

This study utilized a wide-ranging array of functional analyses of mitochondria to uncover the biochemical nature of mitochondrial energetic impairment in skeletal muscle, cardiac muscle, and renal mitochondria in mice with CKD. Although each of the organs/tissues exhibit high levels of energy demand, each demonstrated unique functional characteristics and a differential impact of CKD on mitochondrial OXPHOS. Mitochondrial proteome analyses identify few abundance changes in skeletal muscle mitochondrial, whereas cardiac and renal mitochondrial displayed a modest number of differentially expressed proteins in CKD. These findings shed light on the underlying biochemical changes contributing to altered mitochondrial function and provide a platform for future interventional studies aimed to improve bioenergetics in the CKD milieu.

## Figures and Tables

**Figure 1 cells-10-03282-f001:**
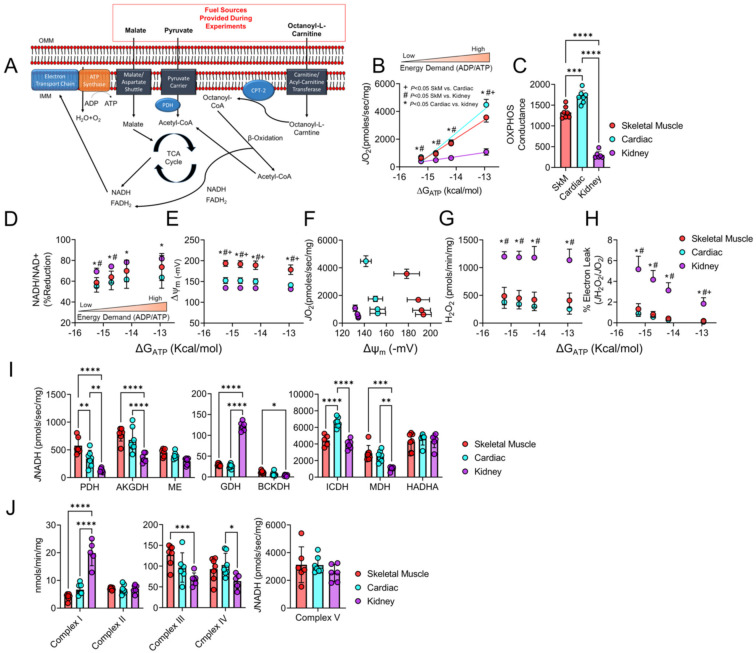
Skeletal muscle, cardiac muscle, and renal tissue mitochondria display distinct bioenergetic profiles in control mice. (**A**) Fuel sources and overall experimental design for bioenergetic assessments. (**B**) Next, a physiological assessment of mitochondrial OXPHOS was employed utilizing a creatine kinase clamp system to mimic a mitochondrial stress test at relevant levels of extramitochondrial energy demand to compare tissue energetics. (**C**) Quantification of each tissue types OXPHOS conductance (slope of *J*O_2_ vs. ΔG_ATP_) (*n* = 6–7/group). (**D**) Relationship between NADH/NAD+ and NADPH/NADP+ redox state expressed as percentage reduction vs. Gibbs energy of ATP hydrolysis (ΔG_ATP_). (**E**) Mitochondrial membrane potential (ΔΨ) in millivolts. (**F**) H_2_O_2_ production vs. ΔG_ATP_. (**G**) Mitochondrial respiratory efficiency evaluated via plotting *J*O2 vs. ΔΨ. (**H**) Mitochondrial electron leak plotted against ΔG_ATP_ (*n* = 6–7/group). (**I**) Matrix dehydrogenase activity was measured for the following enzymes: pyruvate dehydrogenase (PDH), alpha ketoglutarate dehydrogenase (AKGDH), malic enzyme (ME), glutamate dehydrogenase (GDH), branched chain keto acid dehydrogenase (BCKDH), isocitrate dehydrogenase (ICDH), malate dehydrogenase (MDH), and hydroxy-acyl-coA-dehydrogenase (HADHA) (*n* = 6–7/group). (**J**) Enzyme activity of complexes of the electro transport system (ETS) was assessed using standard biochemical assays (*n* = 6–7/group). Data were analyzed using one-way ANOVA and two-way ANOVA followed by Tukey’s post hoc test when an interaction was established. Error bars show standard deviation. *p* < 0.05 *, *p* < 0.01 **, *p* < 0.001 ***, and *p* < 0.0001 ****. (**B**,**D**–**H**) *p* < 0.05 * for cardiac vs. kidney, *p* < 0.05 ^#^ for skeletal muscle vs. kidney, *p* < 0.05 ^+^ for skeletal muscle vs. cardiac.

**Figure 2 cells-10-03282-f002:**
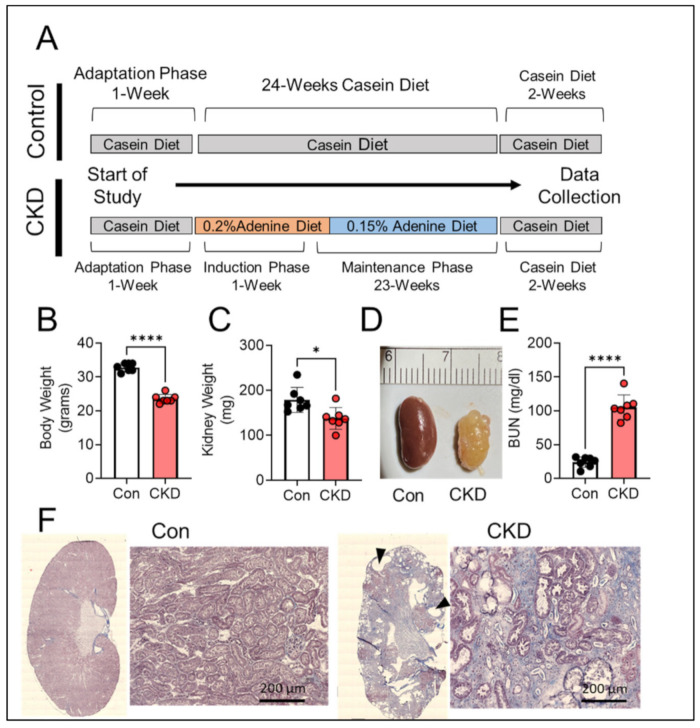
Chronic adenine feeding in mice causes significant kidney pathology. (**A**) Experimental design for diet manipulation for induction of CKD and timing of relevant outcome measures. (**B**) Quantification of body weights in control and CKD mice (*n* = 7/group); (**C**) Quantification of kidney weights of Con vs. CKD mice (*n* = 7/group). (**D**) Representative image of a control and CKD mouse kidney. (**E**) Blood urea nitrogen concentration in the plasma of control and CKD mice measured after 6 months of diet (*n* = 7/group). (**F**) Representative histological image of Masson’s trichrome staining of control and CKD kidney (black triangles identify cyst like structures). Data were analyzed using two-tailed unpaired *t*-test. Error bars show standard deviation. *p* < 0.05 * and *p* < 0.0001 ****.

**Figure 3 cells-10-03282-f003:**
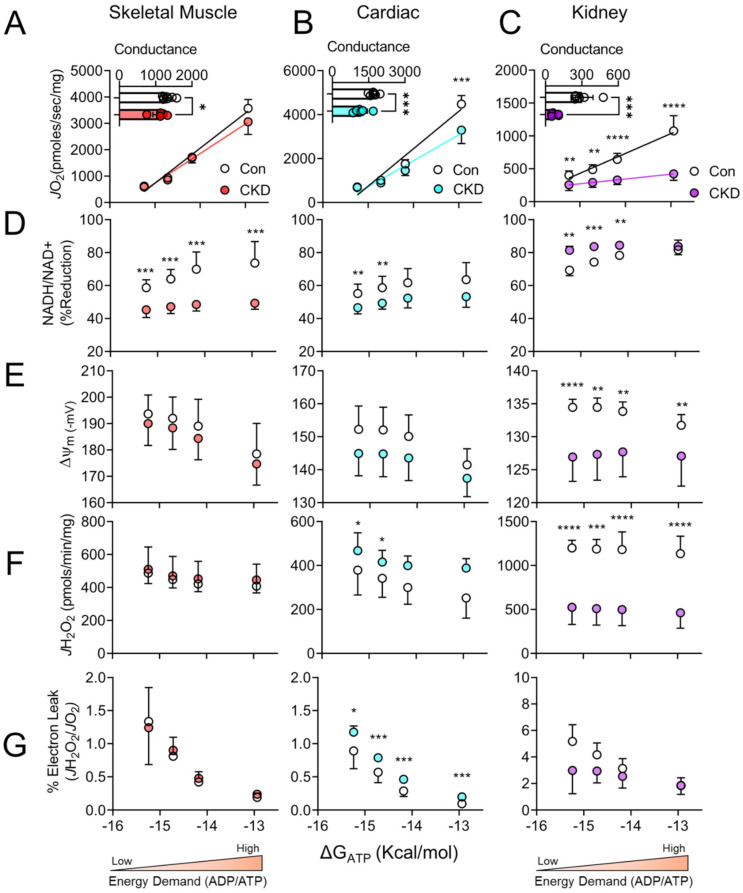
CKD negatively impacts energy transduction in skeletal muscle, cardiac muscle, and renal mitochondria. Mitochondria were isolated from skeletal muscle, cardiac muscle, and the kidney obtained from control and CKD mice. (**A**–**C**) Physiological assessment of mitochondrial OXPHOS was employed to mimic a mitochondrial stress test at relevant levels of extramitochondrial energy demand to compare tissue energetics and conductance was quantified (slope of *J*O_2_ vs. ΔG_ATP_) (*n* = 6–7/group); (**D**) Relationship between NADH/NAD+ and NADPH/NADP+ redox state expressed as percentage reduction vs. Gibbs energy of ATP hydrolysis (ΔG_ATP_); (**E**) mitochondrial membrane potential (ΔΨ) in millivolts vs. ΔG_ATP_; (**F**) H_2_O_2_ production vs. ΔG_ATP_; (**G**) mitochondrial electron leak plotted against ΔG_ATP_ in control and CKD mice (*n* = 6–7/group). Data were analyzed using two-tailed unpaired *t*-test. Error bars show standard deviation. *p* < 0.05 *, *p* < 0.01 **, *p* < 0.001 ***, and *p* < 0.0001 ****.

**Figure 4 cells-10-03282-f004:**
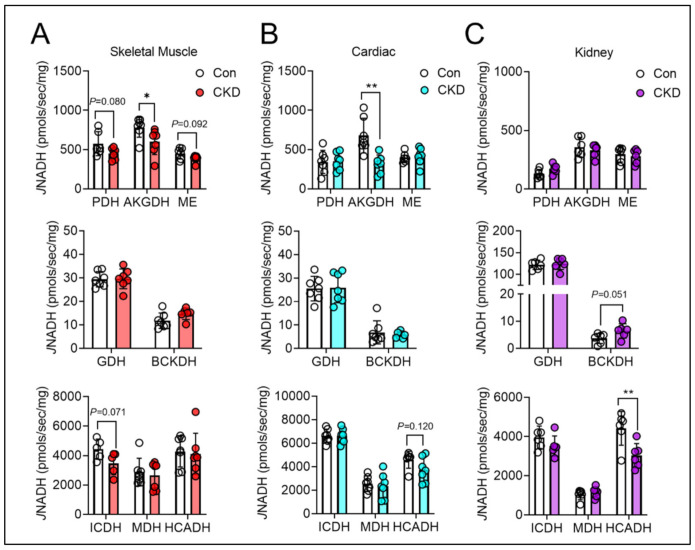
Skeletal muscle, cardiac muscle, and kidney display divergent matrix dehydrogenase activities in CKD. Mitochondria from all three tissues were isolated from control and CKD mice and subsequently permeabilized with alamethicin (0.03 mg/mL) or were lysed in cell lytic and underwent freeze thaw cycles for hydroxy-acyl-coA dehydrogenase (HADHA) assays. (**A**) Matrix dehydrogenase activity measured in skeletal muscle mitochondria; (**B**) cardiac muscle; and (**C**) renal mitochondria of the following enzymes: PDH, AKGDH, ME, GDH, BCKDH, ICDH, MDH, and HADHA (*n* = 6–7/group). Data were analyzed using two-tailed unpaired *t*-test. Error bars show standard deviation. *p* < 0.05 * and *p* < 0.01 **.

**Figure 5 cells-10-03282-f005:**
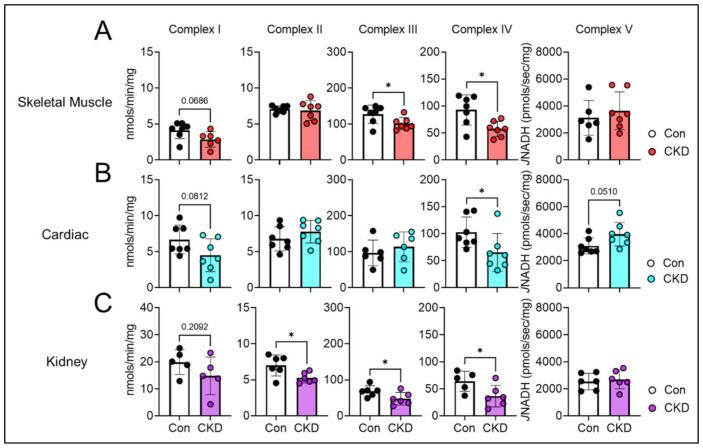
Activity of electron transport system enzyme complexes activity were measured using spectrophotometric biochemical assessments in (**A**) skeletal muscle mitochondria; (**B**) cardiac muscle; and (**C**) renal mitochondria (*n* = 5–7/group). Data were analyzed using two-tailed unpaired *t*-test. Error bars show standard deviation. *p* < 0.05 *.

**Figure 6 cells-10-03282-f006:**
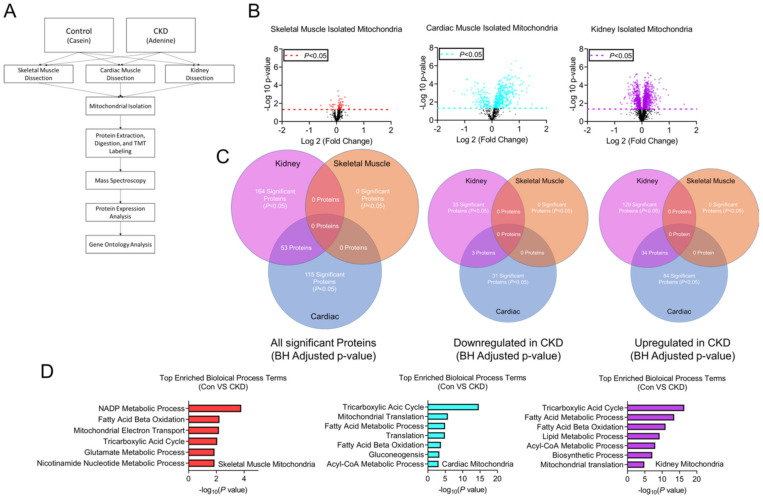
Mitochondria from skeletal muscle, cardiac muscle, and renal tissue express unique mitochondrial proteome alterations in CKD. (**A**) Experimental approach for TMT-labeled proteomic assessments; (**B**) Protein abundance analysis of skeletal muscle, cardiac muscle and renal tissue mitochondria volcano plots using. Dashed lines indicate the unadjusted -log10 *p* = 0.05 demonstrating differentially expressed proteins in CKD mice; (**C**) Venn diagram comparisons of all significantly expressed proteins (adjusted *p* < 0.05), all downregulated proteins, and all upregulated proteins in all three tissues of control and CKD mice; (**D**) Gene ontology analysis of differentially expressed proteins in skeletal muscle, cardiac muscle, and renal tissue isolated mitochondria in control and CKD mice (*n* = 3/group/tissue).

**Figure 7 cells-10-03282-f007:**
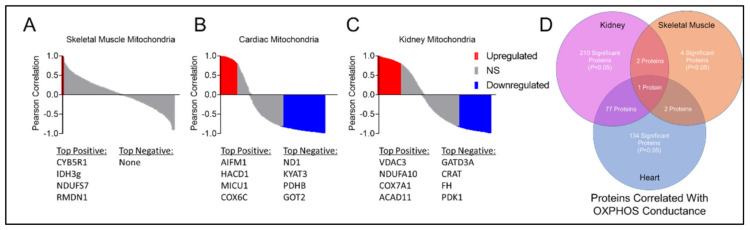
Association between the mitochondrial protein abundance and OXPHOS conductance. Pearson correlations between protein abundance and OXPHOS conductance for (**A**) skeletal muscle; (**B**) cardiac muscle; and (**C**) renal tissue mitochondria in control and CKD mice (*n* = 3/group/tissue). Pearson correlations with *p* < 0.05 are shown in red (positive) and blue (negative). Top positive (red) and negative (blue) correlating proteins are listed below each graph. (**D**) Venn diagram of all the significantly correlated proteins to OXPHOS conductance show common and distinct associations across tissues.

## Data Availability

Proteomics data have been deposited to the jPOST repository (ID JPST000977) and Proteome Xchange (ID PXD021847). All other data associated with this manuscript is available online: https://figshare.com/s/d822931dbb2d52829209 (accessed on 21 November 2021).

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
