# Peer review of "Mitochondrial Bioenergetic and Proteomic Phenotyping Reveals Organ-Specific Consequences of Chronic Kidney Disease in Mice"

_cells, 2021, doi:10.3390/cells10123282_

Round 1

Reviewer 1 Report

In this study, author established a mouse model of CDK, investigated the possible role of mitochondria and the tissue specificity of mitochondria by bioenergetic and proteomic phenotyping of mitochondria from skeletal muscle (SkM), cardiac muscle (CM), and renal tissue from mice with CKD.

  1. Selection of research tissues: Is it more reasonable to select liver for the tissue experiment with higher metabolic level?
  2. The focus of research should be on CKD and attention should be paid to the topic.
  3. Is it unreasonable to use Venn to display the data in Figure 6? Is the form of the list more intuitive?
  4. If more tissue and organ experiments can be added, will the tissue specificity of mitochondria in CKD be more convincing?

Author Response

We thank the reviewer for their helpful suggestions.  Our point-by-point responses are in the attachment below.

Reviewer 2 Report

The article is an interesting and important study of changes in the functioning of mitochondria and changes in their proteome in CKD. It is important that the authors compared several organs, not just the kidneys. The authors compared many parameters of mitochondrial functioning both normally in different organs and against the background of the development of CKD. The article is certainly very important for understanding the role of mitochondria in the pathogenesis of CKD and its effect on other organs, but some issues require clarification.

Comments on the Methods section:

  1. when centrifuging at 800 g, proteins cannot be precipitated, so the phrase "subsequently centrifuged at 800g for 10 minutes to pellet non-mitochondrial proteins…" should be corrected.
  2. The authors do not disclose the method of converting the fluorescence intensity of TMRE into potential values in millivolts. In addition, it is desirable to give (for example, in supplementary material, raw data on measuring the intensity of fluorescence). The authors everywhere give values that are calculated parameters (conductance, delta G, potential) so the protocol of its calculation should be provided.
  3. Also, the protocol for measuring and calculation the activity of individual respiratory chain complexes are not described, since this value is derived from the actual measured parameters, it is also necessary to describe the calculation methodology and provide raw data (at least in the response to the reviewer)

Comments to the results section:

  1. It is not clear in the figures what rest and exercise mean in the diagrams, how it was measured, these concepts are also not indicated anywhere in the text, and they are not discussed
  2. The names of the drawings require correction. Practically, many drawings are called rather methodically "Mitochondria was isolated from healthy control skeletal muscle, cardiac muscle, and renal tissue." Whereas the main name should reflect the essence of what is shown on the figure.
  3. Did the authors evaluate the amount of mitochondria isolated from CKD tissues in comparison with the control? The authors show that the kidney undergoes serious morphological changes. Perhaps the mitochondria in it are severely damaged and their number during isolation could be greatly reduced. Whereas those that were eventually isolated are relatively intact. As a result, we could have a "survivorship bias" error by studying healthy mitochondria and losing damaged ones.

Discussion

The authors do not discuss enough the relationship between changes in the proteome and the detected changes in the functional parameters of mitochondria.

Author Response

We would like to thank Reviewer #2 for their helpful suggestions and comments. The attachment below contains our point-by-point responses and indicates changes to the manuscript accordingly.

Round 2

Reviewer 2 Report

The authors have addressed all my questions